# Application of Multi-Actor Multi-Criteria Analysis for Transition Management in Energy Communities

Maria Luisa Lode *, Geert te Boveldt, Cathy Macharis and Thierry Coosemans

Mobility, Logistics and Automotive Technology Research Center, Vrije Universiteit Brussel, 1050 Ixelles, Belgium; Geert.te.Boveldt@vub.be (G.t.B.); Cathy.Macharis@vub.be (C.M.); thierry.coosemans@vub.be (T.C.)
* Correspondence: Maria.Luisa.Lode@vub.be; Tel.: +32-2-6292286

**Abstract:** Energy communities (ECs) play a role in the transition towards a low-carbon economy by 2050 and receive increasing attention from stakeholders within the energy sector. To foster ECs, transition management (TM) is a promising managerial approach to steer and guide the transition towards more sustainable practices. However, TM lacks a consistent methodology that addresses the criticism of the current application. To investigate what a structured and replicable TM approach for ECs can look like, this paper applies the multi-actor multi-criteria analysis (MAMCA), a participative multi-criteria decision method, to a case study EC in the Netherlands involving various stakeholders. The impact of the application on power relations, the political sphere, sustainability conceptualization, guidance of transitions, and representation was analyzed. MAMCA was found useful for multi-stakeholder settings seen in potential ECs, offering a unifying methodology for the practical application of TM. In the EC setting, the added value of MAMCA within TM lies more in the social representation, insight into stakeholder viewpoints, and communication rather than in final decision-making.

**Keywords:** energy transition; transition management; multi-actor multi-criteria analysis; energy communities; multi-criteria analysis

## 1. Introduction

Although energy communities (ECs) have been legally defined by the European Commission in 2019, the concept remains widely unknown to authorities and energy end-users, and the transposition is still in the beginning [1]. Since 2015, the European Union (EU) has updated and communicated its transition strategy to achieve a net-zero carbon economy by 2050 highlighting that the future energy system will be characterized by decentralization, electrification, liberalization, and integration of energy systems while guaranteeing a fair and just transition that puts citizens at its core [2,3]. The EU considers ECs a potential and holistic solution addressing the social, environmental, technical, institutional, and economic challenges ahead. ECs can be described as integrated community energy systems, which are entities "supplying a local community with its energy requirements from high-efficiency cogeneration or trigeneration energy sources and from renewable energy technologies, coupled with innovative energy storage solutions including the EV [electric vehicles] and energy efficiency demand-side measures" [4].

To foster their development the EU has published two directives, namely the Directive (EU) 2018/2001 on the promotion of the use of energy from renewable sources and the directive (EU) 2019/944 on common rules for the internal market for electricity and amending Directive 2012/27/EU, in which the legal entities of Renewable Energy Community and Citizen Energy Community are created [5,6]. Both the Renewable Energy Community and Citizen Energy Community entities are entitled to participate in the energy sector by producing, consuming, distributing, supplying, and aggregating energy and providing energy services. They are characterized by member participation and by a dedication to

environmental, economic, and social benefits. The member states of the EU must translate these directives into national law and provide enabling frameworks for ECs, but the process is expected to take longer than anticipated [1].

Acceptance of energy generation based on local renewable energy sources such as wind, solar, biomass, geothermal, and hydropower is essential to the success of ECs, since local resistance and the "not in my backyard" resentment can slow down or even stop the deployment of local renewable energy sources [7–9]. Consequently, ECs must be investigated not merely from the perspective of technological advancements, but also under the consideration of the social acceptance of specific EC configurations. The current challenges for the roll-out of ECs call for methodological and practical tools that support the EC development.

Transition management (TM) describes the intentional and structured steering of change towards the desired direction and has found wide-ranged application in cities, districts, and living-labs around the globe [10–13] to foster transitions and the energy transition specifically [14–16]. TM is incorporating methods such as transition pathways and multi-actor views into long-term planning while addressing the social sphere of sustainability [17]. Dóci et al. [18] has used TM as a conceptual and analytical framework to analyze the transformative potential of ECs for the energy market, but the practical implementation of TM for ECs is lacking. Whereas existing work focuses on the theoretical and analytical application of TM, this work looks into the practical application of TM for ECs. In existing case studies, TM was applied using various tools, e.g., multi-actor processes, envisioning, fore- and backcasting, learning, and experimenting [19,20].

Consequently, the existing hands-on applications of TM vary broadly in their usage of methodologies and focus; only a few utilized various tools along the TM cycle, such as Witt et al. [16]. Case studies in which TM was deliberately implemented share most often the following characteristics: An initiator calls for the development of more sustainable practices and mobilizes key stakeholders in the problem setting. Often local, regional, or national government actors take over the role of such an initiator. For instance, the transition towards a more sustainable waste management in the city of Ghent (Belgium) was initiated by local authorities who then reached out to other stakeholders such as citizens, researchers, companies, and other organizations [21]. A similar procedure was implemented in a Finnish case study where five municipalities started to engage with local and national stakeholders to jointly elaborate on potential sustainable tourism visions [22]. In these case studies, exploratory multi-actor processes took place in formal roundtables, interviews, or workshops, but also in informal meetings. In many cases, a TM approach was implemented without an actual intention to apply TM; in the case study of the Cook River in Australia, governance experimentation and the continuous growth of actor involvement and network growth led organically to an TM application [23,24]. While the case studies share prominent elements, methods that guide and allow for scenario evaluation are missing. We observed that a unifying methodology that integrates several of the tools while maintaining the flexibility to different contexts of application is of need.

The multi-actor multi-criteria analysis (MAMCA) was developed by Macharis [25] and belongs to multi-criteria decision analysis methods that allow the inclusion of different criteria from different stakeholders in decision-making and public planning. The methodology was applied in various large transport projects in Belgium [26] but can be applied to a wide range of problems affecting various stakeholders. Thus, we consider MAMCA as a promising method to engage stakeholders to include their criteria into the planning and evaluation of ECs on the local level. Using MAMCA for a TM application is combining scenarios on sustainable trajectories and multi-actor processes with a qualitative evaluation method that fosters learning. We conceive MAMCA as a methodology for practically applying TM. By integrating MAMCA into TM, we examine the applicability of MAMCA. Eemnes, a town in the Netherlands, serves as the case study where the approach was put to test, as it provided a transition environment with high potential for replicability. By applying MAMCA in the case study, we seek to answer the question of how MAMCA can

concretize TM and which advantages and weaknesses result from this approach focusing on the key criticisms on TM. TM and MAMCA will be first explained, and an integrated approach is created. Then the case study design is explained before conclusions on the application and impact on key criticism of TM are drawn.

This work adds to existing research in two ways. Firstly, by contributing to practical TM case studies, and secondly, by expanding participatory tools for the design, implementation, and evaluation of specific EC arrangements through introducing MAMCA.

## 2. Materials and Methods

### 2.1. Transition Management

TM builds on system thinking and a multi-level perspective (MLP) [27–29] in which transitions are analyzed from three different angles: The global, namely "landscapes", which is composed of slow-changing, defining external factors, and well-established actor's interaction, which dictate specific rules and requirements further down the hierarchy in which "regimes", the meso-level, and "niches", the micro-level, are embedded [27]. The meso-level is a composition of various dominant elements within the system, e.g., "technology, infrastructure, industry structure, policy, and techno-scientific knowledge" [27]. The micro-level is conceived, separated, and protected from the dominant system structures. In these niches innovation and novel practices are developed and tested. Niches can carry the potential to replace the dominant regime and create a new equilibrium on the meso-level [29,30].

For the application of TM, transition arenas are formed consisting of a network of actors that experiment with innovations [31]. Transition arenas are located in the niche-level in which "front-runners", such as innovators, developers, and leaders dedicated to foster transition, collaborate to tackle existing socio-technical problems [17]; any of the mentioned stakeholders, policymakers, or academic workers can put TM into practice or initiate it at first instance.

TM is composed of the following elements [17]:

1    Development and set-up of a transition arena
2    Organization of a multi-actor process
3    Delineating the transition problems
4    Development of long-term visions and long-term problem definition
5    Development of transition end-goals that serve as a ground for back-casting methods
6    Exploration of transition pathways
7    Formulation of intermediary goals
8    Agreement on the means and implementation of the goals
9    Evaluation of intermediary goals and lessons learned
10   Iteration

MLP, the conceptualization TM originates from, has been applied both for the analysis and the acceleration of innovation, and its application varies depending on the subject at stake. Foxon et al. [14] identified three main lines of this research: (1) applying it as a framework to analyze transitions from an MLP perspective; (2) it is the conceptualization of transitions on which the application of TM as a practical approach and governance tool is built and which comprehends transitions as a development that can be guided and steered to a certain extent; (3) as a means to develop socio-technical transition pathways that allow looking not only into technology and policy developments, but also into the impacts of the different transition trajectories on the environment, society, and economy.

TM has been closely examined by various researchers highlighting points of caution for its application, especially addressing the lack of drawing a connection to the political sphere of decision making, highlighting points such as power relations, inequality, and social representation [32–34]. This encompasses the conceptualization of sustainability and that desirable long-term goals remain a political prioritization; thus, applying TM and developing transition agendas depend on the parties leading the process [34,35]. Particularly the practical application of TM and the setting of "transition areas" and selection of the

"frontrunners" are targeted by these reflections [33,36]. Reinforcing power relations and their impact on niche developments and uptakes are seen as a key issue of TM applied in theory and practice [10,37–39]. Avelino [37] argues that the existence of political processes within TM cannot be denied; for example, this is shown by the efforts of stakeholders to have and be part of future development pathways. Johnstone and Hielscher [40] added that phasing out non-sustainable practices are part of TM, and Bosman et al. [41] highlight that the new positioning of incumbent stakeholders in new regimes must be considered. Consequently, both the creation of sustainable transition pathways together with destabilizing unsustainable practices must be addressed. Additionally, Hendriks [32] stressed that neglecting the "political arena" [34] in which TM is practiced and overlooking the impact of the selection of key stakeholders is creating a lack of legitimacy with regard to participation, representation, and taking the lead in the transition process [39]. Political steering and support were shown to play a crucial role in the development and success of transition [42].

Building on the mentioned criticism concerning power relations, representation, sustainability conceptualization, and the guidance of the transition process, we investigate if and how the application of MAMCA can complement the critiques of TM and highlight where a need for caution remains.

### 2.2. MAMCA

The MAMCA methodology is based on multi-criteria analysis but adds another layer to the evaluation by incorporating stakeholders. Each stakeholder group has a set of individual criteria that serve as a basis for the evaluation of different scenarios specific to the stakeholders' criteria. MAMCA can be used to conduct the multi-actor processes and long-term planning [26,43] and is tested here in the context of TM and ECs for the first time. For the application of MAMCA, the following steps are identified by Macharis [26].

#### 2.2.1. Defining the Problem and Alternatives:

As a first step, the problem at stake is defined and potential scenarios that depict solutions or alternatives to the problem are constructed. The business-as-usual (BAU) serves as the basis for comparison and represents the current situation. Scenarios can take up very different aspects of a problem such as, among others, long-term policy measures, technical solutions, and organizational structures of a network [26]. To profit most from applying MAMCA, the problem definition and exploration of potential solutions are organized in a participatory way and based on an early-stage involvement of stakeholders [44,45]. Thus, defining the problem and alternative can be conducted simultaneously or in reversed order with the stakeholder identification and engagement.

#### 2.2.2. Stakeholder Identification and Engagement

In this step, people and entities who are affected by or affecting the problem or the consequences of the solution are assessed and consulted to take part in the MAMCA process. This aligns with creating a transition arena and network within TM. Stakeholder consultation is a key part and basis of the other elements of MAMCA. Several authors propose different methods of how stakeholders can be identified; by applying the snowball method in which stakeholders mention whom they consider at stake; based on an iterative approach building on social network theory; or stakeholder theory and salience analysis building on the key characteristics of stakeholders: power, legitimacy, and urgency by [46–48], respectively. MAMCA allows the incorporation of future generations and other social groups that lack legal representation or accessibility in the scenario evaluation.

#### 2.2.3. Criteria Definition and Weighting

Following the participatory approach of MAMCA, stakeholders communicate their criteria, and they can be derived from the problem definition, alternatives, and literature studies. Stakeholders who share the same criteria are clustered together into stakeholder

groups to be homogeneous in their interests [25]. To assess the performance of these criteria in the different scenarios, criteria and key performance indicators are defined. Based on pairwise comparison, e.g., through the analytic hierarchy process (AHP) by Saaty [49], the criteria are ranked according to their importance.

### 2.2.4. Criteria Measurement and Measurement Methods

The previous alternatives to the BAU are then specified further and put into a broader context, leading to more elaborative scenarios [25]. We used the scenarios as boundary objects, allowing us to discuss the problem at stake from different perspectives and with different views on sustainability (e.g., technical vs. social) [50,51]. Scenarios have been a key tool for TM and sustainability research [52]. To evaluate the performance of the criteria within the different scenarios, methods to measure the criteria are set up. Depending on the problem at stake, quantitative and/or qualitative measurement methods are selected and, if necessary, experts are consulted to assess and evaluate the performance of each scenario on each criterion.

### 2.2.5. Overall Analysis and Ranking

Each scenario is evaluated based on its performance on each criterion for each stakeholder group. The input for the evaluation can be submitted by analysts, experts, or the stakeholders themselves. To guarantee a holistic view of the problem, a multidimensional and multidisciplinary group of evaluators should be selected. The input for the evaluation is processed using multi-criteria decision analysis methods such as AHP, PROMOTHEE, MAVT, ELECTRE, MACBETH, or similar, and this results in a ranking of the scenarios [53–55]. The ranking of each scenario for each stakeholder is then visualized in a multi-actor view.

### 2.2.6. Results

The multi-actor overview reveals where opposing views on the scenarios exist. Therefore, apart from the scenario ranking itself, the most important outcome is the insight where stakeholder views are aligned, and where results should be discussed with caution [25]. The results serve as input for discussions and lead the way for further adaptation of the preferred scenario. They enhance the mutual understanding of the present stakeholders and create a sphere of learning [25,26].

### 2.2.7. Implementation and Iteration

If the MAMCA is used as a decision-making tool and one of the scenarios received overall consensus, the needed steps for its implementation are decided upon, and the criteria serve as a reference point to measure progress. Over time, the explained steps can be iterated to optimize and adapt scenarios in response to internal or external changes.

For the EC application, we incorporated MAMCA within TM. In Figure 1, it is shown how we integrated MAMCA within TM: In the outer circle, the TM steps are displayed; in the inner circle, the aligned MAMCA methodology is shown.

The set-up of the transition arena and organization of the multi-actor process within TM corresponds to the MAMCA step to identify the problem at stake and potential alternatives to the current situation. The problem is delineated by the engagement and cooperation with the stakeholders who share and weigh their criteria and communicate their vision of transition end-goals. Long-term visions and potential solutions to the problem are analyzed within MAMCA using scenarios that can be co-developed with the stakeholders. The scenarios represent different pathways to reach specific long-term sustainability goals, e.g., climate neutrality by 2050. These represent the transition pathways within TM. MAMCA allows receiving insight into the internal reasoning of the stakeholders by the criteria and weights they provide as input. Specific performance indicators are formulated to measure the performance of the scenarios, which aligns with the intermediary goal setting within the TM cycle. The performance of the alternatives is evaluated using the proposed methods,

such as the AHP comparison, and indicates which aspects of the transition pathway must be improved. This creates the opportunity to adapt and optimize the selected transition pathway. The criteria serve as a basis to evaluate the progress and performance of the selected scenario towards the defined transition end-goal. The performance indicates how the iteration of problem and scenario development can be improved.

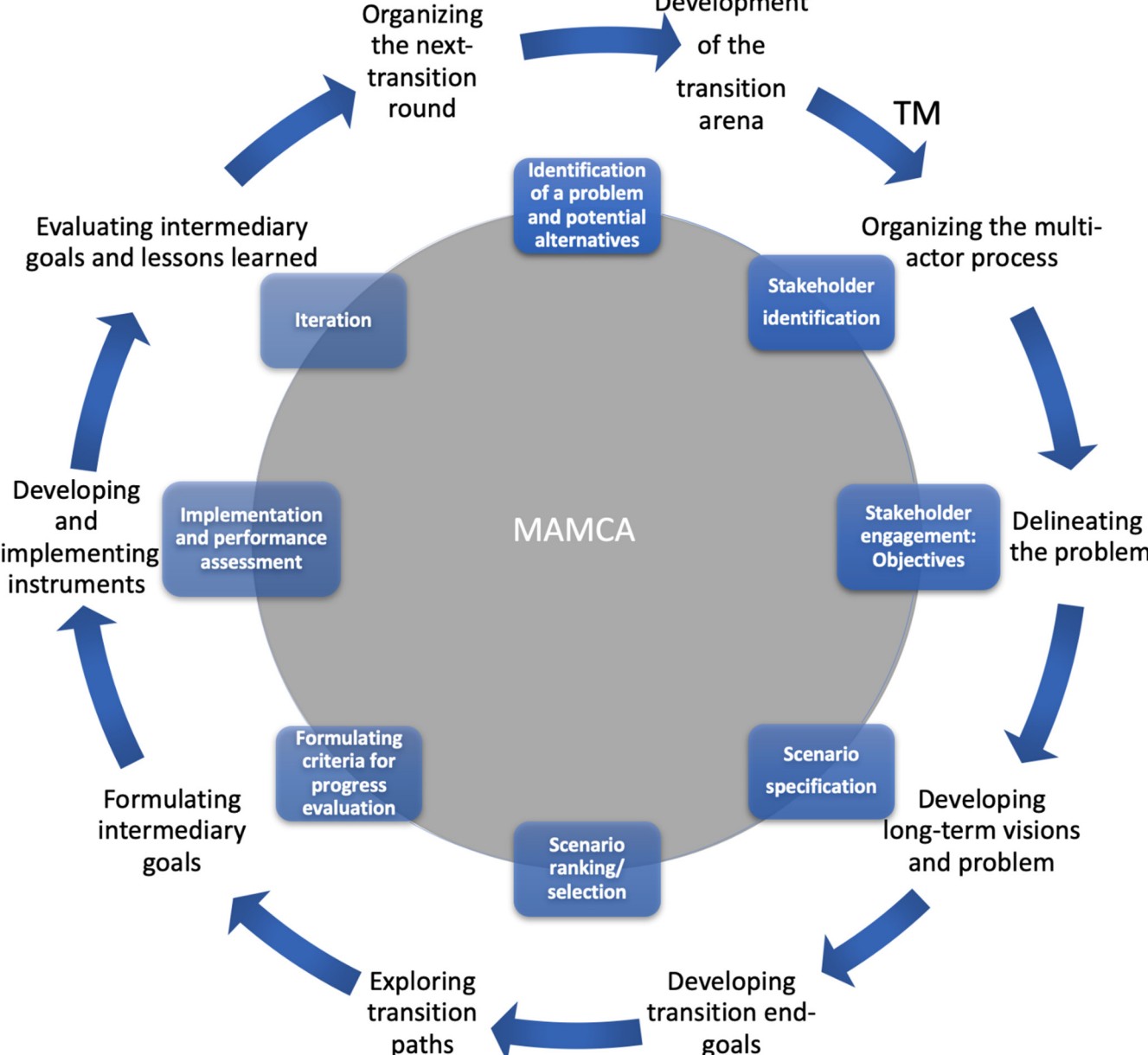

**Figure 1.** MAMCA within the TM cycle developing [31,56].

The integration of MAMCA for a practical TM approach complements each step of the TM with a methodological contribution. It supports TM practitioners with a clear evaluation methodology that suits various problems at stake, such as defining and deciding upon suitable local EC configurations. This facilitates guidance through the TM process as well as provides a mean to engage stakeholders and increase representation. The advantage of using MAMCA is to visualize clearly the opinions and views of different stakeholders, allowing discussions on different transition pathways and sustainability conceptualizations, which are key for social learning and capacity building. Applying MAMCA for TM also supports the cyclic nature of TM by providing a way to monitor progress on the criteria

mentioned. The MAMCA methodology for TM can serve various purposes depending on the context it is applied in: to engage and educate uninformed stakeholders about a problem they are affected by, co-create possible transition pathways, or build capacity in terms of stakeholder networks and informal channels for policy development.

*2.3. Case Study Design*

Eemnes, the case study village in the Netherlands, is part of the EU-funded project RENAISSANCE [57] and was selected for the case study because of its public interest in renewable energy source and the energy transition. Eemnes is composed of 3600 households and has a share of renewable energy source consumption of 12.3%. The primary goal of the project, which runs from 2019–2022, and Eemnes is to foster the energy transition through the increase of renewable energy source share by 20%, end-consumer participation of 200 households, and a decrease of the energy costs for the community members. The transition process at Eemnes was initiated by the municipality that already transitions towards a low-emission energy system shown in a high ratio of individually installed photovoltaic assets on citizens' households as well as by the aspiration of the municipal authority to be climate neutral by 2030. The municipality was granted a regulatory sandbox to experiment with Peer-2-Peer (P2P) energy trading.

Applying MAMCA in this transition context provides a practical example to test the applicability of the approach in general, but also in the specific context of ECs. The case study presents a replicable transition example, indicating which stakeholders can be involved, among others, in local energy markets and how MAMCA allows to engage stakeholders for ECs.

Figure 2 shows how the MAMCA steps were conducted at Eemnes; steps 1 and 2 synchronized in the process. After the consultation and invitation of relevant stakeholders, scenarios were built, stakeholders shared and weighted their criteria, and criteria for evaluation were defined. Based on their evaluation, the multi-actor view was compiled.

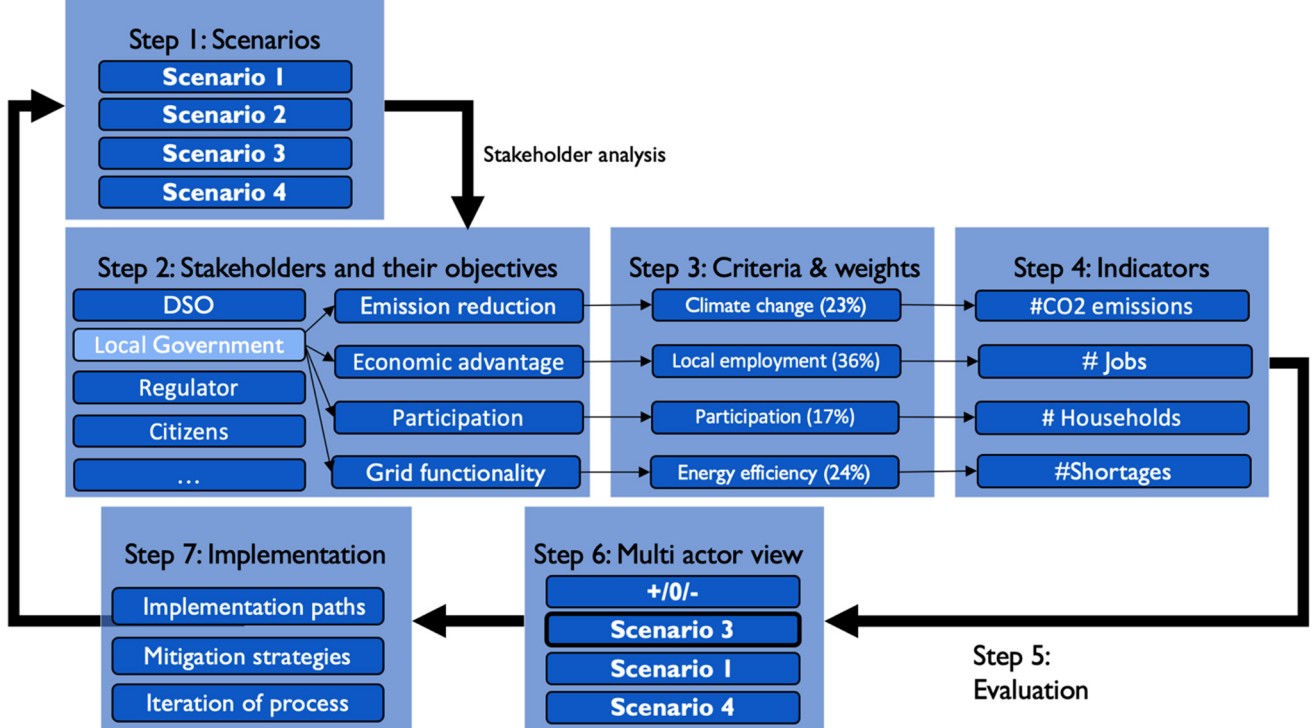

**Figure 2.** MAMCA application at the pilot site, building on [56].

In the case study, the new MAMCA software [58] is used for the explained MAMCA steps to facilitate the collective evaluation, mass participation, and visualization of the different scenarios in a digital manner.

## 3. Results

In the case study, the transition arena was already initiated by a network of progressive stakeholders within the Eemnes municipality. The municipal authority was a driving-force to connect representatives of the civic association and the technology providers for a local energy market and to participate in the project. Before the MAMCA process started, the municipality had decided to become climate-neutral by 2030; the key question they wanted to investigate was how their municipality can transition towards a carbon-neutral and socially accepted integrated community energy system.

Step 1 and 2: Scenario Building and Stakeholder Engagement

Before the stakeholder engagement, the municipality of Eemnes determined for the project that they aim for validation of a local energy market based on flexible prices and with more than 200 households participating in the local market while showcasing lower energy prices for the participants of the EC compared to the current offering from traditional utilities. The following stakeholders were identified and part of the process: representatives of the citizens (EnergieVanNu), local businesses, technology providers (the Platform provider), local authorities, and the distribution system operator (DSO). Instead of an extensive stakeholder mapping, a stakeholder identification based on a snowball method was conducted. Due to time limitations, a compromise in favor of practicability and feasibility was made. In the case study, previous connections were created between the major and the invited stakeholders. The preliminary selection of stakeholders was confirmed by the stakeholders who were mentioned within the snowball method. Inviting regional and national stakeholders can be of advantage, especially if the addressed problem is trans-regional. We set the system boundaries to the geographic boundaries of Eemnes and its local energy system. Depending on the contexts, e.g., where powerful industry representatives are present, remotely located communities are involved and the stakeholders are unclear, a more detailed mapping is advised.

Eemnes developed and explored three different scenarios, namely the BAU (the baseline scenario), the EC, and the aggregation model (AM) scenario, which are described in Table 1. The scenarios mainly differ in their organizational approach and the way energy is purchased and/or traded among different stakeholders.

**Table 1.** Eemnes scenarios.

| Scenario | Description |
|---|---|
| BAU | This reflects the current situation of the pilot site and is the baseline scenario for comparison. Eemnes is composed of 3600 households with a comparably high rate of personal-use renewable energy sources. Eemnes pursues becoming energy neutral by 2030. Surplus energy is directly fed back into the grid and remunerated based on a fixed-feed in tariff. |
| EC | In this scenario, an energy cooperative facilitates the P2P trading mechanism in which all citizens can participate. Demand–respond mechanisms and services can be offered by the EC, and surplus energy could be sold for profit. Storage possibilities are considered as shared investments. Members have decision-making power, and the community is partly independent of the central energy supply. |
| AM | In this scenario, prosumers unite in an exclusive prosumer network independent from their locality but connected digitally. The network acts as a virtual power utility, trading energy while exploiting the options of dynamic pricing and aggregation of energy production. |

One representative of each of the five stakeholder groups was informed about the workshop and project and was asked to fill out a questionnaire. All filled out the survey and shared which criteria apply to their stakeholder group regarding the future energy system and added more to the list where required. The list of the proposed criteria ranged from technical over environmental to social and economic criteria mentioned by energy experts and/or were based on literature findings. The selection of different criteria shows the difference in what the stakeholders consider as important for ECs; this was shown in unique sets of 4–6 criteria per stakeholder group. The respondents were also asked to mention stakeholders that they perceive as having a stake in the problem and problem solution. A snowball method and salience analysis were used to include all stakeholders as proposed by André et al. and Mitchell et al. [48,59]. Before the criteria weighting in the workshop, citizens were asked to share their criteria in addition to EnergieVanNu, and they confirmed the selection of criteria before starting the process.

Step 3: Criteria Weighting

In a community event, citizens and at minimum one representative of each stakeholder group were invited to get introduced to ECs, and an open discussion followed the interactive MAMCA steps. In the workshop, the criteria were weighted, and the scenarios were evaluated by the stakeholders themselves. Figure 3 shows the selected criteria for each stakeholder group (*y*-axis). Using an AHP comparison from a rank of −9 to 9, stakeholders shared how the scenarios score on specific criteria in comparison to each other. They had to compare all criteria, for instance, affordability opposed to grid functionality between −9 and 9, while 0 would denote an indifference, a positive score greater importance of grid functionality, and a negative score greater importance of affordability. A higher score reflects the greater importance of the criteria to the respective stakeholders. The importance of the criteria is displayed by the bars resulting from the AHP comparison of the criteria (*x*-axis). The stakeholders differed in what they consider most important regarding the energy market. EnergieVanNu and the local company placed the most importance on participation, the platform provider on increase in renewable energy sources, the DSO on grid functionality, and the local government on emissions reductions. Step 4 was left out in this overview as the stakeholders conducted the evaluation themselves without using specific data acquisitions or measurements.

Step 5: Scenario Evaluation

In this project, a list of key performance indicators was compiled for each criterion, and they were used to evaluate the performance of the scenarios. For the evaluation, we first asked the stakeholders to evaluate the scenarios themselves by comparing each scenario on the selected criteria. The evaluation was based on an AHP comparison as in the weighting exercise. The stakeholders had to score the perceived impact of the scenarios on all criteria on a −9 to 9 scale. For example, to evaluate the EC scenario, the impact on the criteria reduction of emissions opposed to economic advantage were compared to each other. If the impact on reduction of emissions was considered to be positive, a higher score towards this criterion within the EC scenario was given. The impact of the different scenarios on all criteria was evaluated and resulted in the evaluation scores in the Appendix A.

Step 6: Multi-Actor View

Based on the evaluation, the results for each scenario and stakeholder are displayed. In Figure 4, the multi-actor view is shown resulting from the evaluation by the stakeholders themselves and, therefore, shows subjective views on the scenarios. The BAU scenario is shown in red, the EC scenario in deep blue, and the aggregation model in turquoise. The *x*-axis displays the participating stakeholder groups, and the *y*-axis indicates the value the scenarios have received through the AHP evaluation. The scores of the scenarios cannot be interpreted as an ultimate ranking, because the scenarios are evaluated compared to each other based on the selected criteria of the stakeholders. Therefore, the results

of the evaluation indicate which scenarios are promising or perceived as desirable for the stakeholders concerning the BAU. During the workshop, the multi-actor view was discussed to understand better the reasoning of the weighting and evaluation. The BAU scenario has received lower scores for all participants, indicating that a change to other approaches of the EC or aggregation model is perceived positively. For the DSO, the models score equally the same; during the workshop the representative mentioned, neither of the business designs make a significant difference to the DSO's operation at the site. The aggregation scenario scores better than the BAU scenario, while the EC scenario performs best among all the participants. This strengthens the municipality's decision to further intensify actions towards the implementation of an energy community. Participating citizens showed consensus with the EC scenario, which was aimed to be pursued over the long term.

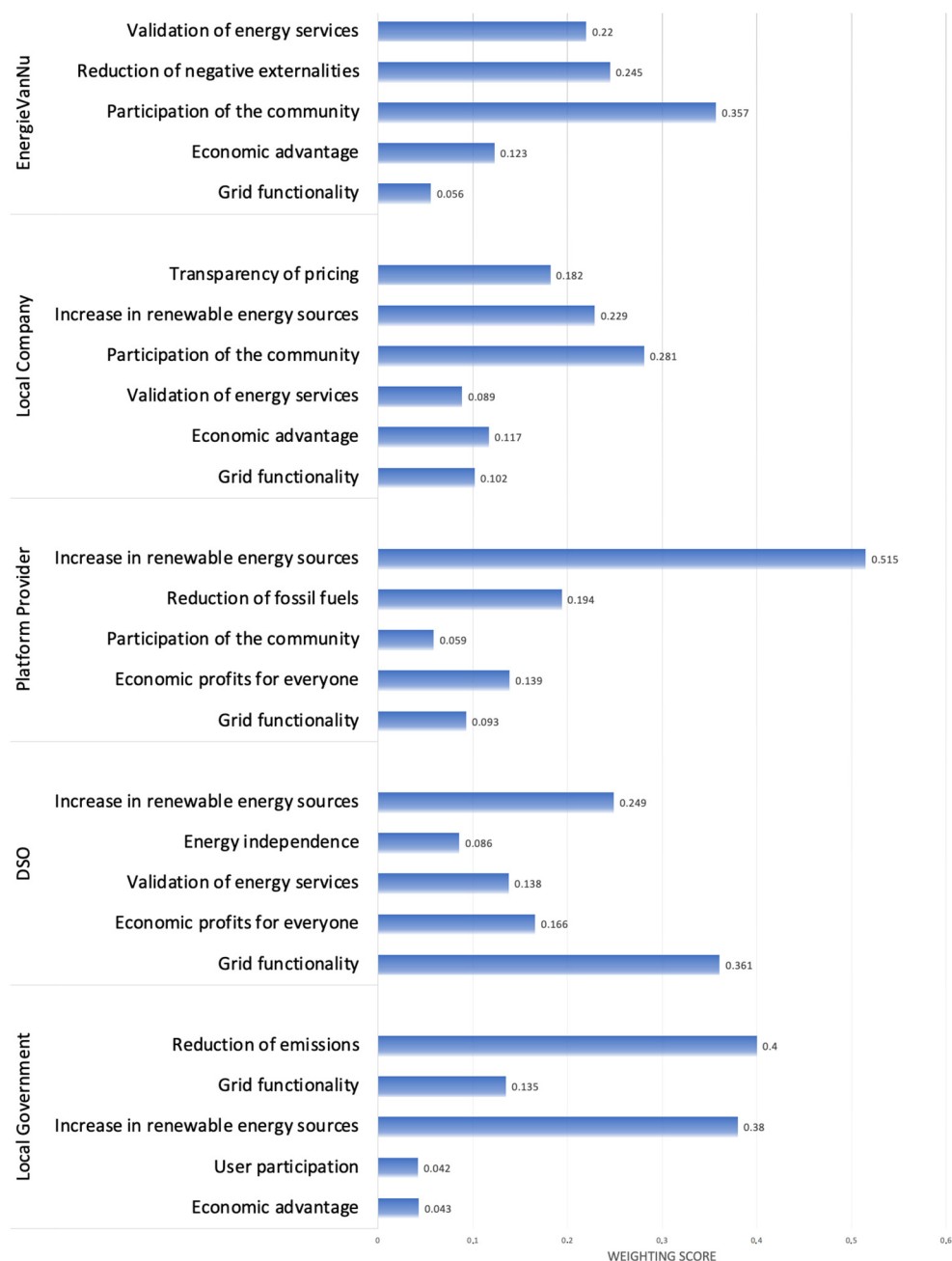

**Figure 3.** Criteria and weighting.

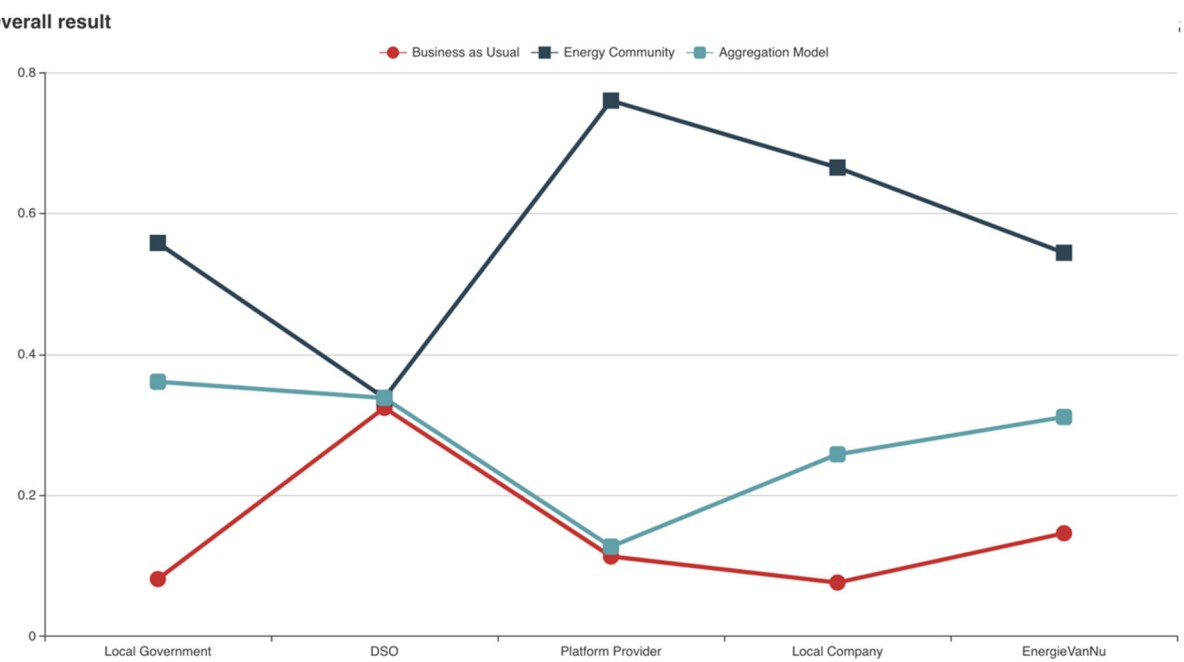

**Figure 4.** Multi-actor view.

Step 7: Implementation and Iteration

With the results and discussion among all stakeholders, the pilot site opted to experiment with the EC scenario and continued the efforts to transition towards a local EC. The selected criteria serve as reference points for future evaluations. A detailed evaluation of technological solutions and organizational configurations based on actual data is possible when the measuring equipment is installed. Following the re-evaluation of the scenarios by experts based on data from the site gives insight into the actual performance of the scenario towards the goal of climate neutrality by 2030. This feedback loop may result in a rephrasing and specification of the scenario.

## 4. Discussion

The case study showed how the MAMCA methodology was incorporated within and for TM. Each step of MAMCA was aligned with the trajectory of the TM cycle. The goal of the MAMCA application was to elaborate with the Eemnes community which EC configurations are possible at the site and which ones are favored by the stakeholders; it also fulfilled an educating purpose.

Building upon the summarized criticism on TM, advantages and disadvantages of applying MAMCA within TM and for ECs are considered.

Guidance of the transition process:

The application of MAMCA and the usage of the MAMCA software structured the interaction with stakeholders during the entire TM process and the stakeholder workshop. The methodology can be applied without using the MAMCA software following the equations provided by Huang et al. [58]. Not using the software can result in increased time investment and the need for the availability of stakeholders who are able to conduct the evaluation. Having non-experienced actors or stakeholders guiding the MAMCA process can be problematic when the guiding stakeholders are not familiar with the methodology and evaluation. The guidance of the MAMCA process was undertaken by academics who were experienced in the MAMCA application and the usage of the software application while not having a stake in the problem setting to prevent unexperienced or incumbent stakeholders taking over the transition process. Addressing the role of incumbent stake-

holders, it is unlikely that incumbent stakeholders who are not interested in a transition would initiate the usage of the MAMCA methodology or TM approach. However researchers take on an important role in sharing knowledge to practitioners and conducting the evaluation [24]. In the case study on the Cook River, authors mentioned a structured approach from the beginning of the engagement process would have helped to pressure the existing regime to transition, and they perceive researchers as one of the key enabling factors for a successful TM application [23]. In TM cases where no academia is involved, the MAMCA software can be of great advantage to conduct the evaluation in a timely manner. As the nature of TM calls for multi-actor processes and researchers, scientists or think tanks were involved in all given examples; the practical application of TM without any of these actors is considered low. Most relevantly, the MAMCA advances TM, representing a tool to replicate the same methodology under different circumstances. It is important to engage most and least powerful stakeholders equally in the process to guarantee fairness. After a network of stakeholders is created, the steps are conducted in a co-creational and multi-stakeholder setting; therefore, the MAMCA methodology should not be attributed to a single stakeholder but to the entire network.

Sustainability conceptualization:

Eemnes had formulated long-term transition goals ahead of the MAMCA application. Taking into account the "X-curve" [60] in which societal transitions are depicted as an interplay and intersection of various streams of different transitions (e.g., technological, institutional, spatial), Eemnes experiments with niches such as ECs while already transitioning towards more sustainable practices. The conceptualization of sustainability in the EC context and the definition of transition pathways were guided and governed by the municipality. While this can be considered problematic concerning the question of who defines sustainability and potential transition pathways, the mediation of the municipality has steered and supported the stakeholder involvement. As indicated by Burke and Stephens [42], political steering on the global and local level was also considered an advantage for the process in this case study. Since citizens and non-energy related stakeholders were not educated on the concept of ECs and the (technical) challenges of the energy sector, the transition pathways for the future integrated community energy systems were only defined by the municipality, with the support of academic actors.

Applying MAMCA allows to define and capture the diversity and variety of ECs. The scenarios used for the MAMCA can take up the different EC characteristics, such as legal and organizational characteristics, sources of renewable energy, number of members, legal restrictions, and costs. MAMCA scenarios can be re-defined during the stakeholder workshop at a stage when stakeholders are more informed about potential scenarios and the local energy transitions as a whole or during a second TM cycle. The top-down sustainability conceptualization by the municipality was considered positive, taking into consideration that scenarios can be adapted at the end of the multi-actor process.

Representation:

MAMCA guides within the "transition arena" but is challenging the concept of "frontrunners" of TM by inviting all stakeholders to the process. In this approach, local policy- and decision-makers were invited to be part of a process that aimed to make decisions with which all stakeholders, also vulnerable societal groups, are content. The MAMCA software simplified the presentation of the multi-actor views, and it allowed for mass participation of citizens. The continuous development of MAMCA and the improvement of its applicability through the supporting software [61] increases its potential for TM with mass participation. Where applicable, e.g., in remote communities, marginalized social groups can be incorporated as separate stakeholder groups to enrich the multi-actor view with more information on demographic characteristics. Representation is considered one of the key advantages of MAMCA; it allows the display of different viewpoints and facilitates the discussion on why specific scenarios are not favored by which stakeholder group. Not consensus itself, but a co-creational approach towards transition pathways and the option to evaluate the impact through qualitative and quantitative data, is an added value to TM.

Nevertheless, representation can only be guaranteed to a certain extent, the risk arises to assume that the opinion in one stakeholder group is homogeneous. This leads over to the disadvantage that the level of representation depends strongly on the process design, time resources, and ability to assume opinions of stakeholder groups that are not able to share their views personally (e.g., future generations, remotely located groups). Using MAMCA facilitates the integration and display of varying opinions, which conclusively results in the discussion of the viewpoints, but does not guarantee a representative viewpoint of all affected parties.

Power relations:

Like the connection between "regime" and "niche" level, the connection between new actors and incumbent actors in the energy sector is also important [42,62]; we argue that the exclusion of incumbent actors is no solution to address power relations in the process of TM. In the case study, the DSO, an incumbent stakeholder, had an indifferent view on the various scenarios discussed. This is likely to be different for other EC settings and environments. In that case, the impact of incumbent stakeholders such as the DSO, transmission system operator, and authorities are expected to have more power for the final implementation of solutions. Subsequently, power relations remain a point of caution. However, ECs are often citizen-led initiatives, and as more energy consumers are becoming prosumers, entities that both produce and consume energy, the insight into their views becomes more important and valuable [63]. A thorough stakeholder identification and stakeholder mapping can serve as a strong base to improve representation and reveal points of caution in case very powerful stakeholders are present. Van Welie et al. [64] have indicated that this is of increased importance in countries in the Global South.

Political sphere:

With the application of MAMCA, the notion of "frontrunners" is extended to inviting a broad range of stakeholders into the TM process. Political decisions that may follow are therewith grounded on stakeholder participation processes. In that way, and by including key authorities in the multi-actor process, the political potential of TM and legitimacy of resulting decisions are strengthened. In the given case study, direct contact with local authorities also facilitated communication with policymakers. This is particularly important for ECs, since adaptations to the legal framework and institutionalization are ongoing.

Adding to the mentioned considerations, we observed three additional concerns. First, the evaluation of scenarios to measure the actual impact on long-term sustainability goals is dependent on skills and resources (e.g., knowledge, time, financial capacities, data). If skills and resources are not available, solutions resulting from uninformed decisions, biases, and a lack of knowledge on the investigated scenarios could be implemented. In practice, this hinders the applicability of the evaluation step of MAMCA, especially when deciding upon technical configurations of ECs, as it requires capacities that might not be available in a remote and economically weak setting. This can also have an impact on the speed of the transition: The application of MAMCA requires time and can slow down the transition process. In the case study, the entire MAMCA process to develop EC scenarios, engage and involve the stakeholders in workshops, and receive their input required one year. The time requirement of a TM approach varies greatly, considering the recursive nature of following a TM approach causes a network to evolve and improves sustainability practices over time. In our case study, solely applying the methodology MAMCA took one year, but implementing TM can spread over decades, such as in the case study on the Cook River [24].

Case studies have shown that TM must be adapted to local requirements, especially when looking into Global South practices [64,65]. The Eurocentric origin story of TM calls for special caution on the possible regime differences around the world, which are characterized by less homogenic and informal structures resulting in a different notion of transition visions [65]. However, as local participation and stakeholder engagement play a key role in this context too, MAMCA can still be applied in an adapted TM context.

Second, a general problem of the approach for pre-defined projects in contrast to "transition arenas" became apparent: key decisions had to be decided ahead to ensure participation in the project; these decisions can then be evaluated with the methodology but not reversed. This traces back to the nature of how projects receive funding guaranteeing the implementation of specific solutions. More differences between pilot projects and transition arenas summarized by van Buuren and Loorbach [66] apply to our case study; stakeholders were involved based on their salience, the project is conducted within existing administrative networks, and the focus lies on creating support and feasibility of the project. In contrast, TM traditionally aims at the implementation of a conscious and intended innovation philosophy outside existing structures with a focus on the creation of a social movement. In the project environment, this can lead to the pitfall that the methodology rather serves as a post-assessment methodology than a co-creational tool.

Third, the focus of this MAMCA application was the development and investigation of new EC configurations, as a result, and connecting it with the 'X-Curve' of transitions [60], more emphasis was placed on experimenting, emergence, and acceleration of EC innovations rather than looking into the institutionalization and stabilization of a new regime. The processes needed to destabilize the unsustainable practices of the current energy market were not taken up in the scenario design. However, TM should take up both aspects by tackling dynamics that support the uptake of sustainable practices simultaneously to the ones that phase-out unsustainable practices. Especially in the context of ECs, the institutionalization of these agonistic and antagonistic dynamics is important and should be considered.

## 5. Conclusions

This work has taken criticism on TM as a starting point to improve the applicability of TM in ECs. With the introduction of the MAMCA methodology within TM and a case study application in the Netherlands, it was shown that the multi-actor process of TM was structured and evaluated in a replicable manner. The application of MAMCA supported the TM approach by combining all relevant elements of the TM cycle in a unifying methodology. This facilitates the application of a TM approach in general problem settings and specifically for the growing roll-out of ECs.

MAMCA opened up the transition arena to non-"frontrunners" and received broad participation of salient and affected stakeholders in the local energy market. By applying the methodology, the connection with the policymakers, represented by the municipal authority, and feedback on the legal barriers that potential EC configurations are faced with, was strengthened. As policymakers are directly involved, decision-making processes can be fastened.

By applying MAMCA, it was possible to display criteria of powerful stakeholders, such as the DSO, as well as underrepresented stakeholders, e.g., energy end-consumers, who all have a stake in future ECs. Involving the stakeholders at the site resulted first in an awareness of the EC concept and then secondly in a discussion based on the different opinions and views of the stakeholders. The MAMCA tool was not used to make a final decision on which EC configuration will be implemented. The added value of the MAMCA methodology to TM is the methodological engagement of various stakeholders according to their stake in the problem and the integration of their criteria from an early stage of a decision process.

In the case study, all stakeholders involved preferred the EC scenario or were not opposing the option to transition to an EC. The current situation was perceived as least favorable compared to the other options, which confirmed that the stakeholders have a desire to transition to a new system setting. For communities that want to explore possible ECs collaboratively or are following a TM approach, MAMCA offers the opportunity to structure this process and to have insight into the different views of the stakeholders and, primarily, educate and discuss the broad concept of ECs.

More broadly, applying the proposed methodology in various EC settings will allow engaging stakeholders to participate, to compare rationales behind different ECs in future research, and can highlight which aspect must be addressed to fasten the transition speed to ECs. Furthermore, the MAMCA tool is not limited to the application of transport or energy-related topics but is proposed as a methodology to implement TM reducing some of the introduced criticism such as social representation, the connection to the political sphere, and sustainability conceptualization.

**Author Contributions:** Conceptualization: C.M. and M.L.L.; methodology: M.L.L.; validation: M.L.L.; formal analysis: M.L.L.; investigation: M.L.L.; writing—original draft preparation: M.L.L.; writing—review and editing: M.L.L. and G.t.B.; visualization: M.L.L.; supervision: G.t.B.; project administration: T.C.; funding acquisition: T.C. All authors have read and agreed to the published version of the manuscript.

**Funding:** This research was funded by the H2020 program, grant number 824342.

**Informed Consent Statement:** As part of the H2020 RENAISSANCE project, this study was approved under consideration of ethics and further aspects.

**Acknowledgments:** The authors would like to thank all involved stakeholders to have participated in the MAMCA process. We would also like to thank the anonymous reviewers for their comments that helped to improve this paper.

**Conflicts of Interest:** The authors declare no conflict of interest. The funders had no role in the design of the study; in the collection, analyses, or interpretation of data; in the writing of the manuscript; or in the decision to publish the results.

## Appendix A

**Table A1.** Stakeholder evaluation (based on the AHP comparison in which all criteria were compared with each other on a −9 to 9 scale, resulting in a score between 0 and 1).

| Local Government | Emission Reductions | Grid Functionality | Increase of RES | User Participation | Economic Advantage | | Score |
|---|---|---|---|---|---|---|---|
| BAU | 0.062 | 0.223 | 0.054 | 0.055 | 0.071 | | 0.081 |
| EC | 0.715 | 0.692 | 0.306 | 0.738 | 0.723 | | 0.558 |
| AM | 0.223 | 0.084 | 0.64 | 0.207 | 0.206 | | 0.361 |
| **DSO** | **Increase RES** | **Energy Independence** | **Validation of Energy Services** | **Economic Advantage** | **Grid Functionality** | | **Score** |
| BAU | 0.111 | 0.143 | 0.143 | 0.143 | 0.667 | | 0.324 |
| EC | 0.444 | 0.429 | 0.429 | 0.429 | 0.167 | | 0.338 |
| AM | 0.444 | 0.429 | 0.429 | 0.429 | 0.167 | | 0.338 |
| **Platform Provider** | **Increase of RES** | **Reduction of Fossil Fuels** | **Participation of the Community** | **Economic Profits for Everyone** | **Grid Functionality** | | **Score** |
| BAU | 0.12 | 0.106 | 0.067 | 0.067 | 0.188 | | 0.113 |
| EC | 0.746 | 0.765 | 0.794 | 0.794 | 0.755 | | 0.760 |
| AM | 0.134 | 0.129 | 0.139 | 0.139 | 0.057 | | 0.127 |
| **Local Company** | **Transparency of Pricing** | **Increase of RES** | **Participation** | **Validation of Energy Services** | **Economic Advantages** | **Grid Functionality** | **Score** |
| BAU | 0.072 | 0.061 | 0.061 | 0.12 | 0.108 | 0.085 | 0.076 |
| EC | 0.673 | 0.723 | 0.723 | 0.331 | 0.624 | 0.701 | 0.665 |
| AM | 0.255 | 0.216 | 0.216 | 0.549 | 0.267 | 0.213 | 0.258 |
| **EnergieVanNu** | **Validation of Energy Services** | **Externalities** | **Participation of the Community** | **Economic Advantage** | **Grid Functionality** | | **Score** |
| BAU | 0.346 | 0.106 | 0.068 | 0.159 | 0.095 | | 0.146 |
| EC | 0.11 | 0.633 | 0.685 | 0.589 | 0.651 | | 0.544 |
| AM | 0.544 | 0.26 | 0.247 | 0.252 | 0.254 | | 0.311 |

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
