# Peer review of "Application of Multi-Actor Multi-Criteria Analysis for Transition Management in Energy Communities"

_sustainability, doi:10.3390/su13041783_

Round 1
Reviewer 1 Report
Overall comments:
- The objective of the study to apply the Multi-Actor Multi-Criteria (MAMCA) tool within the Transition Management (TM) approach to a case study is clear and well described. It is also original in the sense that this has not been done before.
- Figure 1 illustrates the method of the study clearly. A sub-section describing how this new, integrated approach to apply MAMCA within TM can address some of the shortcomings of TM would be good to highlight, what is novel about the research contribution.
- The main weakness of the study is that it is unclear, by who - and how - the MAMCA tool would be applied outside of an academic context. Is it a 'neutral tool' for co-creation of the transition process by multiple stakeholders for multiple purposes or a tool for manipulation of stakeholder inputs by whoever drives the transition and owns the MAMCA software?
- As the authors frame the MAMCA tool as a contribution to practically apply TM and address some of the critiques of this approach regarding power relations, social representation and inequality, the lack of references to other examples of 'how-to' guidance and case studies of applying TM are striking. This includes experience with applying TM in a developing country context and outside of academia such as by the Initiative for Climate Action Transparency (ICAT) Transformational Change Methodology.
Specific comments:
- For reader friendliness avoid acronyms that are not used frequently, for example p. 6 RES, ICES and others. This also goes for the style of referencing using numbers instead of author/date. For instance, p. 3 a sentence starts with a number (33). Starting with a name of the study referenced and bracket the number would read better.
- P. 4 (lines 147-155) and p. 11 it is indicated that the process to apply MAMCA took one year and that the tool is vulnerable to capture by incumbents interested in a particular outcome. Hence, practical guidance on how to use the tool, by who incl. time, resources and skills required, would be useful to help practical application in other contexts outside of academia.
- P. 9, step 5: Further explanation how scenarios are compared based on various stakeholder criteria (Fig. 3, though the text states these are objectives, not criteria, perhaps both?) is needed, as the criteria are not the same for all stakeholders.
- P. 10, Discussion: Is MAMCA software Open Access? If not, what does it take and cost to access it, also in terms of skills and time?
Author Response
Dear Reviewer,
Thank you for your remarks and comments on the submitted paper.
In the following, we would like to explain and highlight how we adapted the paper accordingly.
Are the research design, questions, hypotheses and methods clearly stated?:
In the introduction, a section was added to state more clearly the research design and research question as well as the method that was used. It was justified why a case study approach was opted for (line 97-168).
Applying MAMCA in this transition context provides a practical example to test the applicability of the approach in general but also in the specific context of ECs. The case study presents a replicable transition example indicating which stakeholders can be involved in local energy markets and how MAMCA allows engaging stakeholders for ECs (line 269-272).
Is the article adequately referenced?
We have added more references and information on existing TM case studies. We included case studies in Belgium, Finland and Australia, and indicated specific caution for TM practices in the Global South (the examples summarized findings in Nairobi, Malaysia, Thailand, Tanzania). By doing so, we also tried to answer the request on “How to”-examples while highlighting that MAMCA for TM is such an example. We also highlighted the role of researchers for TM in the given examples (lines 73-88). Response to overall comments:- Figure 1 illustrates the method of the study clearly. A sub-section describing how this new, integrated approach to apply MAMCA within TM can address some of the shortcomings of TM would be good to highlight, what is novel about the research contribution.
The key novelty of using MAMCA for TM is providing a replicable and practical tool for various sustainability problems tackled by a TM approach. A section explaining the advantages concerning the existing shortcomings of TM was added to highlight the advantage of the MAMCA application. Examples on TM practices were added to refer to the current lack of such unifying methodologies that facilitate the practical application of TM. Additionally, MAMCA and TM were not yet conducted in the context of ECs, here the research adds by giving insight into local transition dynamics, local energy actors and their reasoning.
Added line 346-357: The integration of MAMCA for a practical TM approach is novel and complements each step of the TM with a methodological contribution. It supports TM practitioners with a clear evaluation methodology that suits various problems at stake, such as defining and deciding upon suitable local EC configurations in our case study. This facilitates guidance through the TM process as well as provides a means to engage stakeholders and increase representation. The advantage of using MAMCA is to visualize clearly the opinions and views of different stakeholders allowing discussions on different transition pathways and sustainability conceptualizations which are key for social learning and capacity building. Applying MAMCA for TM also supports the cyclic nature of TM by providing a way to monitor progress on the objectives mentioned.
- The main weakness of the study is that it is unclear, by who - and how - the MAMCA tool would be applied outside of an academic context. Is it a 'neutral tool' for co-creation of the transition process by multiple stakeholders for multiple purposes or a tool for manipulation of stakeholder inputs by whoever drives the transition and owns the MAMCA software?
Who: Any societal actor can initiate TM, the added case studies show that most commonly local authorities initiate TM. TM can develop out of an organically grown process of stakeholder involvement and local initiatives. Here, MAMCA helps to crystalize and clarify that a TM approach is followed from an early stage on. After a network of stakeholders was created, the steps are conducted in a co-creational and multi-stakeholder setting, it should not be attributed to a single stakeholder but to a group of stakeholders. In the EC context, MAMCA and TM can be of special interest for citizen cooperatives who want more insight into local needs and requirements. So, even if the TM was initiated by a single stakeholder, the intention is to address complex problem settings that require multi-actor solutions.
These points were added in the paper in the section on “Guidance of the transition process”, line 531-554.
How?: We added examples on practical TM case studies and highlighted the common elements missing of a structured step-by-step implementation/methodology in most cases. It was highlighted that a TM approach is often a result of an organically grown initiative or network which makes it specifically difficult to determine a “how-to” overview. Added lines 74-88
The MAMCA methodology can serve various purposes depending on the context it is applied in; to engage and educate uninformed stakeholders in a problem they are affected by, to co-create possible transition pathways, and to build capacity in terms of stakeholder networks and informal channels for policy development.
In Eemnes, it was used by academia in collaboration with a local authority to educate, inform and engage stakeholders. While final-decision making was not directly impacted by the MAMCA, involved stakeholders and the created network may impact future decisions regarding the EC configuration as TM is a recursive process. Added lines 354-357
Depending on the development stage of an EC, MAMCA can support decisions regarding which EC configuration is suitable in the local context, as it was done in the case study, which joint investments should be made, or if local resistance towards previous decisions occurred MAMCA facilitates understanding the different viewpoints and can lead to more acceptance. Lines added that refer to that: 354-357.
- As the authors frame the MAMCA tool as a contribution to practically apply TM and address some of the critiques of this approach regarding power relations, social representation and inequality, the lack of references to other examples of 'how-to' guidance and case studies of applying TM are striking. This includes experience with applying TM in a developing country context and outside of academia such as by the Initiative for Climate Action Transparency (ICAT) Transformational Change Methodology.
This is an important remark and we added information on the nature of multi-stakeholder processes in TM and on case studies including in the Global South.
The remarks are in the lines: 73-88 (case studies, how-to examples )
Lines: 542-554 ( reflection on academia and involved actors)
Lines 655-660 (caution on Global South)
Response to specific comments:
- For reader friendliness avoid acronyms that are not used frequently, for example p. 6 RES, ICES and others. This also goes for the style of referencing using numbers instead of author/date. For instance, p. 3 a sentence starts with a number (33). Starting with a name of the study referenced and bracket the number would read better.
The following abbreviations are not used anymore: RES, ICES, NIMBY, TSO. We kept the abbreviations where they were used very frequently in the text and where they were needed to understand the inserted figures.
We added the name of the authors where it was needed to guarantee the flow of reading in the text (e.g., every sentence that started with a number or where it was important to differentiate opinions).
- P. 4 (lines 147-155) and p. 11 it is indicated that the process to apply MAMCA took one year and that the tool is vulnerable to capture by incumbents interested in a particular outcome. Hence, practical guidance on how to use the tool, by who incl. time, resources and skills required, would be useful to help practical application in other contexts outside of academia.
We added a section that responses to the time requirements (which vary depending on its context as well as indicated the special role of academia.
See lines: 650-655 and 542-549.
- P. 9, step 5: Further explanation of how scenarios are compared based on various stakeholder criteria (Fig. 3, though the text states these are objectives, not criteria, perhaps both?) is needed, as the criteria are not the same for all stakeholders.
It was indeed written not clearly, we decided to only use "criteria" for this case study as both objectives and criteria aligned in this study.
- 10, Discussion: Is MAMCA software Open Access? If not, what does it take and cost to access it, also in terms of skills and time?
I hope we could answer and respond to your comments sufficiently and we thank you again for the review.
Reviewer 2 Report
Taking criticism of TM as a springboard this research seeks to increase the applicability of TM for energy communities. Utilising MAMCA with a case study goes a good distance to highlight how processes related to actors can be appreciated in a better manner using this novel approach, and also generating a route in which the research could be replicated.
The application of MAMCA in this manner gives insight into objectives of stakeholders, offering a view as to how an EC could evolve into the future. This view of stakeholders and their role into the future also gives rise to offering a platform to discussions on how varying opinions and views need to be incorporated into decision making to best configure ECs. This hopefully then provides a route in which to generate early stage engagement.
Methodologically, the research offers a good representation of the routes used to generate data, however the value of the case study needs to be better discussed, and the route in which stakeholder identification took place also needs to be enhanced.
In terms of stakeholder mapping, additionally the rigour of this stage needs to be deepened to ensure replicability of the study is possible.
With ECs being so diverse and varied in nature, there also needs to be a discussion as to how this approach tackles this heterogeneity.
Author Response
Dear Reviewer,
Thank you for your remarks and comments on the submitted paper.
In the following, we would like to explain and highlight how we adapted the paper accordingly.
- Is the content succinctly described and contextualized with respect to the previous and present theoretical background and empirical research (if applicable) on the topic?
We have added more references and information on existing TM case studies. We included case studies in Belgium, Finland, and Australia, and indicated specific caution for TM practices in the Global South (the examples summarized findings in Nairobi, Malaysia, Thailand, Tanzania) (lines 73-88).
- Are the research design, questions, hypotheses and methods clearly stated?:
In the introduction, a section was added to state more clearly the research design and research question as well as the method that was used. It was justified why a case study approach was opted for (line 97-168).
Applying MAMCA in this transition context provides a practical example to test the applicability of the approach in general but also in the specific context of ECs. The case study presents a replicable transition example indicating which stakeholders can be involved in local energy markets and how MAMCA allows engaging stakeholders for ECs (line 269-272).
- For empirical research, are the results clearly presented?
We have not changed the existing figures and also have not added new figures in the paper. All data that was produced was included in the paper. But to understand the visualization of the research better, we added more explanation in the evaluation step. We also indicated clearly that the evaluation scores are found in the annex and added remarks on the software evaluation.
Line:476-487 (evaluation) and 532-537 (software evaluation)
- Is the article adequately referenced?
We have added more references and information on existing TM case studies. We also highlighted the role of researchers for TM in the given examples (lines 73-88). Furthermore, we adapted the existing reference style and corrected where the name of the author of the studies was missing. Most prominently in the introduction lines: 215-234 and lines 540-554
-
Methodologically, the research offers a good representation of the routes used to generate data, however, the value of the case study needs to be better discussed, and the route in which stakeholder identification took place also needs to be enhanced.
We agree with the comment and added in the introduction: "
Eemnes, a town in the Netherlands, serves as the case study where the approach was put to test as it provided a transition environment with a high potential for replicability. By applying MAMCA in the case study we seek to answer the question how MAMCA can concretize TM and which advantages and weaknesses result from this approach focusing on the key criticisms on TM. TM and MAMCA will be first explained and an integrated approach is created. Then the case study design is explained before conclusions on the application and impact on key criticism of TM are drawn." (line 97-168) More specification was also done in the lines 346-357, and 371-374.
To specify more clearly how the stakeholders were identified in the case study, we added information to Step 1 and Step 2.
Added lines 417-425:
"Instead of an extensive stakeholder mapping, a stakeholder identification based on a snowball method was conducted. Due to time limitations, a compromise in favor of practicability and feasibility was made. In the case study, previous connections were created between the major and the invited stakeholders. The preliminary selection of stakeholders was confirmed by the stakeholders who were mentioned within the snowball method. Inviting regional and national stakeholders can be of advantage, especially if the addressed problem is trans-regional. We set the system boundaries to the geographic boundaries of Eemnes and its local energy system. Depending on the contexts, e.g., where powerful industry representatives are present, remotely located communities are involved and the stakeholders are unclear, a more detailed mapping is advised."
- In terms of stakeholder mapping, additionally, the rigor of this stage needs to be deepened to ensure replicability of the study is possible.
We aimed to respond to this remark with the previously mentioned section. Additionally, we made clear that in our case study an existing network was created which resulted in less time investment of this stage for our case. Applicable ways to conduct a stakeholder analysis/mapping are provided in lines: 259-263. We also highlighted points of caution for other contexts in line: 655-660. Practicability and accuracy of the stakeholder analysis can be conflicting in regard to the multi-actor process, to make all identified stakeholders participate in the process is not often feasible.
- With ECs being so diverse and varied in nature, there also needs to be a discussion as to how this approach tackles this heterogeneity.
Lines 587 -590:
"Applying MAMCA allows to define and capture the diversity and variety of ECs. The scenarios used for the MAMCA can take up the different EC characteristics, such as legal and organizational characteristics, sources of renewable energy, number of members, legal restrictions and costs."
We consider this aspect one of the advantages of MAMCA as it allows us to grasp the diversity of ECs in the scenarios that are being compared to the current situation.
I hope we could answer and respond to your comments sufficiently and we thank you again for the review.
Round 2
Reviewer 2 Report
Revisions have been carried out well; specifically to more clearly identify how stakeholders were identified in the case study.
Also in terms of stakeholder mapping, the detailing of how an existing network was created to gain recipients is well outlined. Along with stakeholder mapping discussion and areas of caution detailed.
The author has also done well to thoroughly cover how the proposed technique deal with the heterogeneous nature of ECs.